# Rates, causes, place and predictors of mortality in adults with intellectual disabilities with and without Down syndrome: cohort study with record linkage

Sally-Ann Cooper ,[1] Linda Allan,[1] Nicola Greenlaw,[2] Paula McSkimming,[2] Adam Jasilek,[2] Angela Henderson ,[1] Colin McCowan,[3] Deborah Kinnear,[1] Craig Melville[1]

[1]Institute of Health and Wellbeing, University of Glasgow, Glasgow, UK
[2]Robertson Centre for Biostatistics, Institute of Health and Wellbeing, University of Glasgow, Glasgow, UK
[3]School of Medicine, University of St. Andrews, Aberdeen, UK

**Correspondence to**
Sally-Ann Cooper;
Sally-Ann.Cooper@glasgow. ac.uk

## ABSTRACT

**Objectives** To investigate mortality in adults with intellectual disabilities: rates, causes, place, demographic and clinical predictors.
**Design** Cohort study with record linkage to death data.
**Setting** General community.
**Participants** 961/1023 (94%) adults (16–83 years; mean=44.1 years; 54.6% male) with intellectual disabilities, clinically examined in 2001–2004; subsequently record-linked to their National Health Service number, allowing linkage to death certificate data, 2018.
**Outcome measures** Standardised mortality ratios (SMRs), underlying and all contributing causes of death, avoidable deaths, place, and demographic and clinical predictors of death.
**Results** 294/961 (30.6%) had died; 64/179 (35.8%) with Down syndrome, 230/783 (29.4%) without Down syndrome. SMR overall=2.24 (1.98, 2.49); Down syndrome adults=5.28 (3.98, 6.57), adults without Down syndrome=1.93 (1.68, 2.18); male=1.69 (1.42, 1.95), female=3.48 (2.90, 4.06). SMRs decreased as age increased. More severe intellectual disabilities increased SMR, but ability was not retained in the multivariable model. SMRs were higher for most International Statistical Classification of Diseases and Related Health Problems, 10th Revision chapters. For adults without Down syndrome, aspiration/reflux/choking and respiratory infection were the the most common underlying causes of mortality; for Down syndrome adults 'Down syndrome', and dementia were most common. Amenable deaths (29.8%) were double that in the general population (14%); 60.3% died in hospital. Mortality risk related to percutaneous endoscopic gastrostomy/tube fed, Down syndrome, diabetes, lower respiratory tract infection at cohort-entry, smoking, epilepsy, hearing impairment, increasing number of prescribed drugs, increasing age. Bowel incontinence reduced mortality risk.
**Conclusions** Adults with intellectual disabilities with and without Down syndrome have different SMRs and causes of death which should be separately reported. Both die younger, from different causes than other people. Some mortality risks are similar to other people, with earlier

## Strengths and limitations of this study

► Thorough methods of case ascertainment for intellectual disabilities at baseline.
► Individual verification of intellectual disabilities and its severity, and detailed health assessments at baseline.
► Longitudinal design.
► Large cohort size and study duration, and successful record linkage for 94% of participants.
► Limitations include that the study was conducted in only one part of Scotland, and the reliance on recorded cause of death from death certificates.

mortality reflecting more multimorbidity; amenable deaths are also common. This should inform actions to reduce early mortality, for example, training to avoid aspiration/ choking, pain identification to address problems before they are advanced, and reasonable adjustments to improve healthcare quality.

## INTRODUCTION

People with intellectual disabilities die at a younger age than other people; on average, 20 years younger,[1] or 28 years specifically for people with Down syndrome.[2] It has been demonstrated that people with intellectual disabilities receive poorer management of their long-term conditions within primary healthcare services compared with the general population,[3] and it is conceivable that this is one contributor to earlier mortality. It has been suggested that as many as 40% of deaths of people with intellectual disabilities may have been amenable to good quality healthcare.[4–6] There has been a recent increase in research on mortality in people with intellectual disabilities, but very little research has distinguished people with

intellectual disabilities with and without Down syndrome, or investigated the factors associated with risk of mortality, and causes of mortality.

Previous studies on death in people with intellectual disabilities had limitations such as small sample sizes, or non-representative populations. More recently, there have been large-scale studies which are more representative, having been drawn from intellectual disabilities registers, or social security or primary care data with record linkage to death certification. These have been undertaken in parts of Sweden, Australia, England, Finland, Canada, Ireland and USA(online supplementary table 1).[5–19] These studies fairly consistently report standardised mortality ratios (SMRs) to be high for people with intellectual disabilities, more so at younger ages and higher for women than men. Adult studies have tended to report SMRs in the region of 2–4, although in some, SMR is only slightly above 1.[10 16 19] However, direct comparison between studies is not always possible, due to the different age ranges studied and methods of reporting.

In view of the methods that studies have used for population identification (typically, routine administrative data linked to death certifications), they provide little information on the socioclinical factors that influence SMR, or the risk factors associated with death, beyond that of age and sex. Three studies reported SMR by level of intellectual disabilities, with, broadly speaking, higher SMR with more severe intellectual disabilities.[7 10 17] Only three studies (different studies to those that reported on level of intellectual disabilities) were able to report data separately for adults with intellectual disabilities with and without Down syndrome; two found higher mortality rates for adults with Down syndrome (SMR=7.6[9] and HR=9.21[5]) than for adults without Down syndrome, or an OR showing Down syndrome as a risk of death.[12] A further study reported SMR=5.5 for children and adults (combined) with Down syndrome, but did not report SMR for those with intellectual disabilities without Down syndrome.[20] Two studies reported adults with intellectual disabilities to have higher SMRs if they have the comorbidities of epilepsy,[5 7] and cerebral palsy,[7] as opposed to not having these comorbidities. One study reported adults with intellectual disabilities with comorbid autism to have lower risk of mortality than those without comorbid autism.[5] One study reported the risk factors for mortality in a population with intellectual disabilities to be age, Down syndrome, cerebral palsy, blindness/low vision, technological dependence/medical fragility, wheelchair dependence, mobility impairment without wheelchair dependence, and epilepsy.[12] Factors not found to be risks, if any, were not reported, and a further limitation was that factors were reported by agency staff, rather than the individuals undergoing health assessments.[12] We have not identified any other studies that investigated risk factors for time to mortality in adults with intellectual disabilities.

There is less consistency regarding the most common certified underlying causes of death in adults with intellectual disabilities, partly as some studies do not report these separately for children and adults, or by age ranges. Additionally, studies group causes of death in different ways (eg, pneumonia vs respiratory system), which can affect prevalence rankings between studies. Pneumonia, other respiratory diseases and diseases of the nervous system were reported to be the most common in one study,[11] diseases of the circulatory system and respiratory systems in another,[5] heart disease, neoplasm and Alzheimer disease in a third,[18] and diseases of the circulatory system, neoplasm and the nervous system in a fourth.[19] In adults with intellectual disabilities, cause-specific SMRs have been reported to be high across most groups of disorders.[5 11] These studies did not report cause of death separately for adults with and without Down syndrome. Given the different health profile of people with Down syndrome compared with people with intellectual disabilities of other causes, this is an important limitation.[21] In people with Down syndrome, most studies on mortality have been conducted with child populations, and report the most common causes of death to be congenital heart disease, and pneumonia/diseases of the respiratory system.[2]

Overall, the existing body of literature on mortality in adults with intellectual disabilities does not include more detailed information on level of intellectual disabilities, nor separate out the population with, from those without, Down syndrome (for whom causes of death may differ), nor investigate health and demographic predictors of death other than age and sex, and is inconsistent with regard to causes of death. A better understanding of these factors may provide a pathway to action to reduce the observed earlier mortality in adults with intellectual disabilities.

This study aims to investigate the rates, causes, place and demographic and clinical associations with mortality in adults with intellectual disabilities, with and without Down syndrome.

## METHODS
### Participants
The adult (aged 16+ years) intellectual disabilities population living within the NHS Greater Glasgow area was identified through multiple sources between 2000 and 2001. General practitioners were financially incentivised to identify their registered patients with intellectual disabilities, and all 631 (100%) did so. Adults were also identified via the intellectual disabilities health and social work services including day services, the Health Board register and records of financial payments for any service by social work. This process led initially to an overidentification, such as people with IQ scores in the 70–80 range with additional complex health needs. All were systematically reviewed by nurses in the intellectual disabilities health service, and this group were removed. Thus, a register was compiled, and subsequently updated annually via general practices, with central support from the intellectual disabilities health service, until 2017 when

services were redesigned. The identified adult prevalence of intellectual disabilities within the area was 3.33 per 1000 in 2000–2001.

## Process and data collection

With initial piloting in 2001, each participant had a detailed assessment of their general and mental health, and demographic factors, completed 2002–2004. One of the six specially trained, registered nurses reviewed each person's primary health care records, then used a semi-structured tool, the C21st Health Check, to assess clinical factors and the level and cause of intellectual disabilities. In addition to a review of existing health problems and all bodily health systems, a physical examination was undertaken, including assessment of vision and hearing, measurement of height and weight and a phlebotomy protocol followed. All information was then reviewed by the nurse with one of three general practitioners with a special interest in intellectual disabilities, and any further investigations that were indicated were completed. Previously known, and newly identified, conditions were then classified using the *International Statistical Classification of Diseases and Related Health Problems, 10th Revision (ICD-10)*.[22] Anyone identified to have possible, probable or definite mental ill-health, autism or problem behaviours was then fully assessed by the project's intellectual disabilities psychiatrists. Each person's assessment findings were then case conferenced by the two Consultant psychiatrists, and diagnoses were derived and agreed according to clinical diagnoses, *ICD-10 (Diagnostic Criteria for Research)*,[23] *Diagnostic and Statistical Manuel of Mental Disorders-IV-TR*[24] and *Diagnostic Criteria for Psychiatric Disorders for use with Adults with Learning Disabilities*.[25] Information was also collected on demographics, and community, hospital, and social service use. Further details are provided elsewhere.[26 27] The data were entered into a database by two dedicated data-entry staff.

Each person in Scotland is given a number unique to them at birth or first registration with a general practitioner, which is used in almost all subsequent health service encounters, and on certification of death. The numbers are held on the Community Health Index (CHI) database at National Services Scotland. These CHI numbers provided a means to record link each participant with National Records for Scotland death certification data. This linkage was performed in 2018, and the linked data were held in the NHS Greater Glasgow & Clyde (NHS GG&C) Safe Haven. Data on immediate, underlying and contributory causes of deaths by ICD-10 codes, age at death and place of death were extracted.

In order to provide finer granularity of cause of death, two clinical academics then grouped specific causes of death into narrower groupings than those provided by ICD-10 chapter headings (online supplementary table 2). This approach was also in view of the recognised issue of variation between health staff in distinguishing and recording immediate causes of death, and because some causes occurred in low numbers so could not be

individually reported due to the risk of statistical disclosure. Additionally, some conditions likely to be the same are spilt between different ICD-10 chapters, for example, dementia in Alzheimer disease (F00) and unspecified dementia (F03) in the ICD-10 mental and behavioural disorders chapter, and Alzheimer's disease (G30) and Alzheimer's disease, unspecified (G30.9) in the ICD-10 diseases of the nervous system chapter. A list of related conditions was generated by one of the clinical academics and then checked by the second.

## Analyses

All statistical analyses were conducted using R for Windows V.3.3.0 or SAS V.9.3 and were performed within the NHS GG&C Safe Haven environment. Due to disclosure principles of the Safe Haven, results with counts of less than five cannot be released; these have been referred to as <5 throughout. Similarly, if it is deemed possible that participants may be identified from the results, these may be omitted. Details are provided if this occurred.

Data were summarised for the population of adults aged 16+ years with intellectual disabilities. Categorical variables were summarised with the number and percentages of people falling into each category and the number of missing data. Continuous variables were summarised with the number of observations and those missing, the mean and SD, and the minimum and maximum values, unless otherwise stated.

Participant characteristics were summarised overall and for those alive and those deceased. For those who are deceased, their data including age at death, underlying/contributing causes of death, and location of death were summarised for those with and without Down syndrome. Location codes for place of death are provided where available. We assumed those with the code for non-institutional location to have died at home. Due to small numbers, location codes have been grouped together for NHS hospitals, home, and other hospitals/care facilities including hospices.

Mortality incidence rates have been calculated using the number of deaths in the cohort divided by the number of person years alive within the study period multiplied by 100 000, overall and for those with and without Down syndrome. SMRs were calculated using population data for those aged 15 and over within NHS GG&C in 2010.[28 29] Death rates for males and females by 5-year band ages groups spanning from 15 to 20 years old to 90 years and over were summed to form the expected death rates for the general population. The observed death rate for adults with intellectual disabilities was taken from our study results. The observed/expected death rates were calculated for the intellectual disabilities cohort overall then separately by age group, sex, ability level, and for the adults with and without Down syndrome, and ICD-10 chapter for cause of death, and compared with the general population.

Deaths were also analysed for those that could be considered as deaths that would have been avoidable.

The Office for National Statistics (ONS) published a definition of avoidable mortality,[30] which lists the causes of amenable deaths (deaths that should not occur in the presence of good healthcare, eg, respiratory disease) and causes of preventable deaths (eg, from diseases that could have been avoided by prior immunisation), by ICD-10 codes. Causes of death for the adults with intellectual disabilities have been summarised by ONS definition of avoidable deaths.

To determine the demographic and clinical factors associated with death in adults with intellectual disabilities, time to event analyses were explored using univariate Cox Proportional Hazards models. Variables were selected as potentially relevant on the basis of what is known on causes of death in people with intellectual disabilities, the 20 most common physical health conditions reported in the adult population with intellectual disabilities,[21] and other factors hypothesised as potentially clinically relevant (online supplementary table 3):

▶ Demographics—nine variables.
▶ Clinical conditions—33 variables.
▶ Service use—three variables.
▶ Prescriptions—five variables.

All 50 variables were then permitted entry in to a single multivariable analysis using stepwise regression methods, in order to identify a model containing the statistically significant factors associated with death. Age at date of the health assessment was entered into the model as a continuous measure. Results from the univariate Cox Proportional Hazards models (online supplementary table 3) and the statistically significant multivariable model from the stepwise results have been presented with HRs with corresponding 95% CIs (HR, 95% CI) and p-values were obtained.

### Patient and public involvement

This study was designed to respond to the growing concern expressed by people with intellectual disabilities, their families and third sector organisations about the early deaths of people with intellectual disabilities. The Scottish Learning Disabilities Observatory, where this research was undertaken, has a specific remit for people with intellectual disabilities. Its steering group includes partners from third sector organisations, including Down syndrome Scotland, and people with intellectual disabilities, who approved the work plan for this project prior to it commencing. Results from this study will be disseminated for people with intellectual disabilities in an easy-read version via the Scottish Learning Disabilities Observatory.

## RESULTS
### Population characteristics

Of note, 962 of the original 1023 (94.0%) adults with intellectual disabilities who were assessed were linked to a CHI number enabling the extraction of relevant death data. Reasons for the unlinked 61 people could be due to moving out of the area, or a recording mistake. One further participant was removed from the analysis due to inaccurate recording of dates, leaving 961 adults in the cohort (93.9%). Of these 961 adults, 294 (30.6%) had a record of death. Table 1 shows the baseline characteristics of the full cohort of 961, the adults who died and those still alive at the time of linkage.

### Age at death and mortality incidence

The mean age at death was 61.0 years (SD=7.0 years). Of the 961 adults, 64 (35.8%) of the 179 adults with Down syndrome and 230 (29.4%) of the 782 adults without Down syndrome had a record of death. Their mean age of death was 56.9 years (SD=4.3 years) for the adults with Down syndrome, and 62.2 years (SD=7.5 years) for the adults without Down syndrome. Mortality incidence for the cohort during the study period was 3049.0 per 100 000 person years follow-up, with 3832.1 per 100 000 for those with Down Syndrome and 2885.0 for those without Down syndrome.

### Standardised mortality ratios

Compared with the general population, the SMR was 2.24 (1.98, 2.49) overall; 5.28 (3.98, 6.57) for adults with Down syndrome, 1.93 (1.68, 2.18) for adults without Down syndrome; 1.69 (1.42, 1.95) for men and 3.48 (2.90, 4.06) for women. SMRs were higher the more severe the level of intellectual disabilities, with people with profound intellectual disabilities having an SMR of 4.14 (3.11, 5.17). SMR was high for all age groups (though for the 15–25 year age group, the wide CI includes one, perhaps due to the smaller number of deaths in this group); this decreased as age increased. SMRs were high for most ICD-10 chapter groups of conditions, particularly so for congenital malformations at 17.26 (10.75, 23.78), diseases of the digestive system at 16.13 (8.23, 24.04), mental and behavioural disorders at 12.64 (3.27, 22.00) and external causes at 11.08 (3.40, 18.76). Details are shown in table 2.

### Causes of death

Cause of death data was available from death certificates for 262 (89.1%) of 294 participants who had died, which include 57 (89.1%) participants with Down syndrome, and 205 (88.7%) participants without Down syndrome. Table 3 shows the underlying causes of death by ICD-10 chapters separately for the adults with and without Down syndrome. For the whole cohort, diseases of the respiratory system were the most common (21.8%), then diseases of the circulatory system (19.1%), then diseases of the nervous system (13.0%) and neoplasms, followed by congenital anomalies (10.3%). For the adults with Down syndrome, congenital anomalies were the most common (in all cases this was a record of 'Down syndrome'), then jointly diseases of the respiratory system and diseases of the circulatory system, then diseases of the nervous system, followed by infections, and mental and behavioural disorders. For the adults without Down syndrome, diseases of the respiratory system were the most common, then diseases of the circulatory system,

**Table 1** Cohort characteristics at time of the health assessment, summarised overall and by death status during the follow-up period

| Variable | Statistics/groups | All participants (n=961) | Deceased participants (n=294) | Alive participants (n=667) |
|---|---|---|---|---|
| Age (years) | Mean (SD) | 44.1 (14.6) | 52.4 (13.6) | 40.5 (13.6) |
| | Min, max | 16–83 | 18–83 | 16–77 |
| Age group | 16–25 years | 127 (13.2%) | 10 (3.4%) | 117 (17.5%) |
| | 26–35 years | 153 (15.9%) | 26 (8.8%) | 127 (19.0%) |
| | 36–45 years | 246 (25.6%) | 49 (16.7%) | 197 (29.5) |
| | 46–55 years | 205 (21.3%) | 85 (28.8%) | 120 (18.0%) |
| | >55 years | 230 (23.9%) | 124 (42.0%) | 106 (15.9%) |
| Sex | Male | 525 (54.6%) | 154 (52.4%) | 371 (55.6%) |
| | Female | 436 (45.3%) | 140 (47.5%) | 296 (44.4%) |
| Ability level | Mild ID | 382 (39.7%) | 92 (31.2%) | 290 (43.5%) |
| | Moderate ID | 236 (24.5%) | 73 (24.7%) | 163 (24.4%) |
| | Severe ID | 180 (18.7%) | 67 (22.7%) | 113 (16.9%) |
| | Profound ID | 163 (17.0%) | 62 (21.1%) | 101 (15.1%) |
| Accommodation type | Family carer | 374 (38.9%) | 70 (23.8%) | 304 (45.6%) |
| | Independent | 93 (9.7%) | 36 (12.2%) | 57 (8.5%) |
| | Paid support | 435 (45.2%) | 161 (54.6%) | 274 (41.1%) |
| | Congregate care | 59 (6.1%) | 27 (9.2%) | 32 (4.8%) |
| Down syndrome | No | 782 (81.4%) | 230 (78.2%) | 552 (82.8%) |
| | Yes | 179 (18.6%) | 64 (21.7%) | 115 (17.2%) |

ID, intellectual disabilities.

then neoplasms, then diseases of the nervous system, followed by diseases of the digestive system. Table 4 presents the most common underlying causes of death by individual causes, or related groups of causes, with finer granularity than ICD-10 chapter headings (groups are shown in online supplementary table 2). Causes are listed in the order of how common they were in the whole cohort. Data are presented separately for the adults with and without Down syndrome. For the whole cohort, the most common cause was aspiration/reflux/choking, then respiratory infection, then other malignancy (non-gastrointestinal), then other condition (mostly unrelated conditions that could not be reported individually or as groups, due to individually occurring at a frequency of <5). For the adults with Down syndrome, Down syndrome was the most common cause, then dementia, then other infection. For the adults without Down syndrome, aspiration/reflux/choking was the most common cause, then respiratory infection, then other malignancy (non-gastrointestinal). For the 21 people whose death certificate listed Down syndrome as their underlying cause of death, the death certificates were reviewed and underlying cause of death reclassified, as a sensitivity check. Following this, the most common underlying causes of death for the adults with Down syndrome were dementia (n=20; 35.1%), then other infection (n=7; 12.3%).

Table 5 shows the all contributing causes of death data, again presenting the most common causes by individual causes, or related groups of causes with finer granularity than ICD-10 chapter headings. Data are presented separately for the adults with and without Down syndrome. For the whole cohort, respiratory infection was the most common cause (27.1%), followed by aspiration/reflux/choking (19.8%), other conditions (15.6%), other cardiovascular conditions (non-acute myocardial nor other ischaemic heart disease: 14.5%), then other respiratory conditions. For the adults with Down syndrome, Down syndrome was the most common, then dementia, then respiratory infection, then aspiration/reflux/choking. For the adults without Down syndrome, respiratory infection was the most common cause, then aspiration/reflux/choking, then other condition, then other respiratory conditions and intellectual disabilities.

### Avoidable deaths
According to the ONS list of avoidable deaths, 102 (38.9%) of the 262 deaths were avoidable; most notably, respiratory infection and epilepsies (table 4); 78 (29.8%) were deaths that are amenable to good healthcare, while 51 (19.5%) were preventable deaths, and 27 (10.3%) deaths were classed as both amenable and preventable deaths. This compares to published Scottish death data showing in 2018 that 28% of deaths were avoidable; 14%

**Table 2** Standardised mortality ratios

| Variable | Groups | SMR (95% CI) |
|---|---|---|
| All participants | – | 2.24 (1.99 to 2.50) |
| Age group* | 15–25 years | 18.73 (0.37 to 37.09) |
| | 26–35 years | 4.21 (1.29 to 7.13) |
| | 36–45 years | 3.86 (2.28 to 5.44) |
| | 46–55 years | 3.77 (2.90 to 4.74) |
| | >55 years | 1.86 (1.60 to 2.12) |
| Sex | Male | 1.69 (1.42 to 1.95) |
| | Female | 3.48 (2.90 to 4.06) |
| Ability level | Mild ID | 1.60 (1.27 to 1.92) |
| | Moderate ID | 2.10 (1.62 to 2.58) |
| | Severe ID | 2.78 (2.11 to 3.44) |
| | Profound ID | 4.14 (3.11 to 5.17) |
| Down syndrome | No | 1.93 (1.68 to 2.18) |
| | Yes | 5.28 (3.98 to 6.57) |
| Underlying causes of death grouped by ICD-10 chapter† | Congenital malformations, deformations and chromosomal abnormalities | 17.26 (10.75 to 23.78) |
| | Diseases of the blood and blood-forming organs and certain disorders involving the immune mechanism | 7.50 (-7.20 to 22.20) |
| | Diseases of the circulatory system | 5.55 (4.01 to 7.09) |
| | Diseases of the digestive system | 16.13 (8.23 to 24.04) |
| | Diseases of the genitourinary system | 3.65 (0.73 to 6.57) |
| | Diseases of the musculoskeletal system and connective tissue | 5.40 (-0.71 to 11.52) |
| | Diseases of the nervous system | 7.73 (5.13 to 10.32) |
| | Diseases of the respiratory system | 6.78 (5.02 to 8.54) |
| | Diseases of the skin and subcutaneous tissue | 2.75 (–2.64 to 8.15) |
| | Endocrine, nutritional and metabolic diseases | 3.43 (1.05 to 5.81) |
| | External causes of morbidity and mobility | 11.08 (3.40 to 18.76) |
| | Infectious and parasitic diseases | 8.93 (1.78 to 16.07) |
| | Mental and behavioural disorders | 12.64 (3.27 to 22.00) |
| | Neoplasms | 6.31 (4.19 to 8.43) |
| | Symptoms, signs and abnormal clinical and laboratory findings, not elsewhere classified | 19.51 (0.39 to 38.63) |

*Data used for comparison with General Population (GG&C Health Board) provide data in 5-year age bands therefore 15+. Data on adults with ID are 16+.
†Negative Lower CI and wide CIs indicate low number of observed deaths in study population.
ICD-10, International Statistical Classification of Diseases and Related Health Problems, 10th Revision; ID, intellectual disabilities; SMR, standardised mortality ratios.

amenable, and 24% preventable, similar to the figures in the previous 4 years (data not available prior to 2014).[31] For the 57 deaths of adults with Down syndrome, 17 (29.8%) deaths were avoidable, 15 (26.3%) deaths were amenable to good healthcare, while seven (12.3%) were preventable, and five (8.8%) were both amenable and preventable. For the 205 deaths of adults without Down syndrome, 85 (41.5%) were avoidable, 63 (30.7%) deaths were amenable to good healthcare, while 44 (21.5%) were preventable, and 22 (10.7%) were both amenable and preventable.

### Place of death

Of the 262 participants for whom place of death was known, 158 (60.3%) died in an NHS Hospital, 70 (26.7%) died at home, and 34 (13.0%) died within other hospitals/care facilities. This was similar for both the adults with Down syndrome: 31 (54.4%) in an NHS hospital, 17 (29.8%) at home, and nine (15.8%) within other hospitals/care facilities; and the adults without Down syndrome: 127 (62.0%) in an NHS hospital, 53 (25.9%) at home, and 25 (12.2%) within other hospitals/care facilities.

AUTHOR PROOF

**Table 3** Underlying causes of death grouped by ICD-10 chapter, where cause of death is known

| ICD-10 chapter | Participants with Down syndrome (n=57) | Participants without Down syndrome (n=205) |
|---|---|---|
| Certain infectious and parasitic diseases | 5 (8.8%) | <5 |
| Neoplasms | <5 | 33 (16.1%) |
| Diseases of the blood and blood-forming organs and certain disorders involving the immune mechanism | <5 | <5 |
| Endocrine, nutritional and metabolic diseases | <5 | 8 (3.9%) |
| Mental and behavioural disorders | 5 (8.8%) | <5 |
| Diseases of the nervous system | 7 (12.3%) | 27 (13.2%) |
| Diseases of the eye and adnexa | <5 | <5 |
| Diseases of the ear and mastoid process | <5 | <5 |
| Diseases of the circulatory system | 8 (14.0%) | 42 (20.5%) |
| Diseases of the respiratory system | 8 (14.0%) | 49 (23.9%) |
| Diseases of the digestive system | <5 | 16 (7.8%) |
| Diseases of the skin and subcutaneous tissue | <5 | <5 |
| Diseases of the musculoskeletal system and connective tissue | <5 | <5 |
| Diseases of the genitourinary system | <5 | 5 (2.4%) |
| Pregnancy, childbirth and the puerperium | <5 | <5 |
| Certain conditions originating in the perinatal period | <5 | <5 |
| Congenital malformations, deformations and chromosomal abnormalities | 21 (36.8%) | 6 (2.9%) |
| Symptoms, signs and abnormal clinical and laboratory findings, not elsewhere classified | <5 | <5 |
| External causes of morbidity and mortality | <5 | 7 (3.4%) |
| All deaths | 57 (100%) | 205 (100%) |

ICD-10, International Statistical Classification of Diseases and Related Health Problems, 10th Revision.

## Factors associated with risk of death

The results from the univariate cox proportional hazards models indicated that of the original 50 potential variables, factors associated with risk of death were (online supplementary table 3) as follows:

**Table 4** Underlying causes of death grouped by specific individual causes or related groups of causes, where cause of death is known

| Causes | Participants with Down syndrome (n=57) | Participants without Down syndrome (n=205) |
|---|---|---|
| Aspiration/reflux/choking | <5 | 22 (10.8%) |
| Respiratory infection | <5 | 21 (10.3%) |
| Down syndrome | 21 (36.8%) | <5 |
| Other malignancy | <5 | 19 (9.3%) |
| Other condition | <5 | 17 (8.3%) |
| Epilepsies | <5 | 13 (6.4%) |
| Acute myocardial infarction | <5 | 13 (6.4%) |
| Gastrointestinal malignancy | <5 | 12 (5.9%) |
| Stroke | <5 | 11 (5.4%) |
| Other cardiovascular disease | <5 | 11 (5.4%) |
| Other respiratory condition | <5 | 9 (4.4%) |
| Other infection | 5 (8.8%) | 6 (2.9%) |
| Cerebral palsy | <5 | 11 (5.4%) |
| Dementia | 9 (15.8%) | <5 |
| Other gastrointestinal disorders | <5 | 8 (3.9%) |
| Ulcer/gastrointestinal perforation | <5 | 7 (3.4%) |
| Diabetes | <5 | 7 (3.4%) |
| Other congenital condition | <5 | 6 (2.9%) |
| Other ischaemic heart condition | <5 | 6 (2.9) |
| Mental health | <5 | <5 |
| Other neurological conditions | <5 | <5 |
| Renal failure | <5 | <5 |
| All deaths | 57 (100%) | 205 (100%) |

► Demographics–age at the time of the health assessment, more severe learning disabilities, accommodation type (not living with family carer), not having day-time occupation, and being a smoker (but not sex, the extent of neighbourhood deprivation, civil status, nor Down syndrome, in view of the CIs).

► Clinical conditions–having spastic quadriplegia, hearing impairment, visual impairment, diabetes, percutaneous endoscopic gastrostomy/tube fed, constipation, ataxia/gait disorder, osteoporosis, hypertension, dysphagia, dyspnoea, gastro-oesophageal reflux disorder, lower respiratory tract infection, total number of physical health disorders, not having

**Table 5** All contributing causes of death grouped by specific individual causes or related groups of causes, where cause of death is known

| Causes | Participants with Down syndrome (n=57) | Participants without Down syndrome (n=205) |
|---|---|---|
| Respiratory infection | 22 (38.6%) | 49 (23.9%) |
| Aspiration/reflux/choking | 11 (19.3%) | 41 (20.0%) |
| Down syndrome | 43 (75.4%) | <5 |
| Other condition | 8 (14.0%) | 33 (16.1%) |
| Other cardiovascular disease | 8 (14.0%) | 30 (14.6%) |
| Other respiratory conditions | <5 | 31 (15.1%) |
| Other infection | 9 (15.8%) | 24 (11.7%) |
| Intellectual disabilities | <5 | 31 (15.1%) |
| Epilepsies | 8 (14.0%) | 24 (11.7%) |
| Dementia | 24 (42.1%) | <5 |
| Other neoplasms | <5 | 23 (11.2%) |
| Cerebral palsy | <5 | 24 (11.7%) |
| Acute myocardial infarction | 5 (8.8%) | 19 (9.3%) |
| Other gastrointestinal disorders | <5 | 18 (8.8%) |
| Diabetes | <5 | 19 (9.3%) |
| Other ischaemic heart disease | <5 | 19 (9.3%) |
| Renal failure | <5 | 16 (7.8%) |
| Stroke | <5 | 17 (8.3%) |
| Other congenital condition | <5 | 15 (7.3%) |
| Gastrointestinal malignant neoplasm | <5 | 12 (5.9%) |
| Ulcer/gastrointestinal perforation | <5 | 10 (4.9%) |
| Mental health | <5 | 10 (4.9%) |
| Other neurological condition | <5 | 8 (3.9%) |
| Heart failure | <5 | 7 (3.4%) |
| Injuries and accidents | <5 | 8 (3.9%) |
| Medical/surgical complications | <5 | <5 |
| Secondary malignancies | <5 | <5 |
| Thyroid disorders | <5 | <5 |
| Metabolic disorder | <5 | <5 |
| All deaths | 57 (100%) | 205 (100%) |

**Table 6** Multivariable model results for the outcome time to death

| Variable | HR | 95% CI | P |
|---|---|---|---|
| Age at time of health assessment | 1.056 | 1.046 to 1.066 | <0.0001 |
| Smoker | | | |
| No | 1 | – | |
| Yes | 1.531 | 1.1011 to 2.128 | 0.0112 |
| Down syndrome | | | |
| No | 1 | – | |
| Yes | 2.44 | 1.787 to 3.332 | <0.0001 |
| Epilepsy | | | |
| No | 1 | – | |
| Yes | 1.511 | 1.173 to 1.946 | 0.0014 |
| Hearing impairment | | | |
| No | 1 | – | |
| Yes | 1.32 | 1.030 to 1.692 | 0.0284 |
| Bowel incontinence | | | |
| No | 1 | – | |
| Yes | 0.49 | 0.376 to 0.640 | <0.0001 |
| Diabetes | | | |
| No | 1 | – | |
| Yes | 2.346 | 1.553 to 3.542 | <0.0001 |
| PEG/tube fed | | | |
| No | 1 | – | |
| Yes | 2.346 | 1.135 to 5.989 | 0.0024 |
| Lower respiratory track infection | | | |
| No | 1 | – | |
| Yes | 1.782 | 1.315 to 2.415 | 0.0002 |
| Total number of prescribed drugs | 1.066 | 1.016 to 1.118 | 0.0085 |

PEG, percutaneous endoscopic gastrostomy.

behaviour, eating disorder including pica, nor any mental illness).

▶ Service use—number of general practitioner consultations in the previous 12 months, total number of different types of health professionals providing care at the time of the clinical assessment (but not number of accident and emergency attendances in the previous 12 months).

▶ Prescriptions—antiepileptic drugs, total number of different types of drugs (but not antipsychotic drugs, antidepressant drugs, nor anxiolytic drugs).

Table 6 shows the final model of the variables retained in the multivariable analysis for time to death. The significant factors indicating an increased risk of death were increased age at the time of the health assessment, smoking, Down syndrome, diabetes, being percutaneous endoscopic gastrostomy/tube fed, lower respiratory tract infection at cohort inception, epilepsy, hearing

impaired mobility, not having urinary incontinence, not having bowel incontinence, and not having autism (but not epilepsy, body mass index, nail disorder, epidermal thickening, cerebral palsy, fungal infection, musculoskeletal pain, bone deformity, dental/oral problem, eczema/dermatitis, psychosis, affective disorder including bipolar affective disorder, problem

impairment, and total number of different types of drugs prescribed, while bowel incontinence showed a reduced risk of death. Of note, level of intellectual disabilities, while significant in the univariate analysis, was not retained in the multivariable model.

## DISCUSSION
### Principle findings and interpretation

As far as we are aware, this is the first population-based study of adults with intellectual disabilities to report in detail the factors associated with time to death, and to describe their causes of death, and quantify the SMR separately for adults with and without Down syndrome. This is important, since adults with Down syndrome form a notable proportion of all adults with intellectual disabilities (19% in this cohort), and because they have a different pattern of clinical conditions compared with other adults with intellectual disabilities.[21] We found that aspiration/reflux/choking is the most common underlying cause of death in adults with intellectual disabilities, followed by respiratory infection. They are also the most common all contributing causes of death. The profile differed in the adults with Down syndrome for whom 'Down syndrome', followed by dementia, were recorded as the most common underlying cause of death, and all contributing causes of death (or alternatively, dementia, then other infection were the most common underlying causes when 'Down syndrome' deaths were reclassified); with the next most common all contributing cause of death being respiratory infection, then aspiration/reflux/choking. The proportion of deaths that would have been amenable to good care for adults with intellectual disabilities was more than double that seen in the general population. Although aspiration/reflux/choking is not included in the ONS list of avoidable deaths, and therefore not included in the figures we report on amenable deaths, we consider that good care also could have prevented many of these aspiration/reflux/choking deaths. This appears to be very important for adults with intellectual disabilities irrespective of whether they have Down syndrome. Similarly, some other causes of deaths within this cohort (online supplementary table 2), such as constipation/mega-colon, and urinary tract infections do not appear on the ONS list of avoidable deaths.

Clearly, this pattern of causes of death differs from that seen in the general population, in whom the most common underlying causes of death are heart disease, then dementia, then lung cancer in men, and dementia, then heart disease, then stroke in women.[32] When all cancers are grouped together, in the general population, cancer is the leading underlying cause of death in 30% of men and 26% of women, compared with this study reporting 0% for adults with Down syndrome, and 15.2% for adults with intellectual disabilities without Down syndrome—presumably as the adults with intellectual disabilities are dying younger from other causes, and cancers increase with age.

We found an overall SMR of 2.24; 5.28 in the adults with Down syndrome and 1.93 for the adults without Down syndrome. SMRs were higher for most ICD-10 chapter groupings of conditions. It was higher in the women than the men, as has been previously reported in most (online supplementary table 1), but not all[10 19] previous reports. The reason for this is unknown; in the general population, mortality rates have fallen in recent decades, and more so in middle and older aged men than women (ie, the sex gap is narrowing at these ages), but we do not know what trends over time there have been for people with intellectual disabilities. Having intellectual disabilities removes differences in lifespan by sex compared with the general population; but sex was not a predictor of mortality in our study, so the SMR difference may only be because of the difference found in the general population by sex. SMRs were lowest with older age groups, likely to be due to increased illness in the older general population and conversely a healthier group with intellectual disabilities living to older ages compared with those who die younger (as has previously been reported).[33] Although SMR was higher with increasing severity of intellectual disabilities, ability level was not retained within the multivariable model on time to death. The factors that were independently associated with increased risk of death, in order, were being percutaneous endoscopic gastrostomy/tube fed, Down syndrome, diabetes, having a lower respiratory tract infection at entry to the cohort, smoking, epilepsy, hearing impairment, total number of prescribed drugs and age, while bowel incontinence had a reduced risk of death. Some of these predictors are similar to those reported in the general population, suggesting that earlier mortality of adults with intellectual disabilities is largely accounted for by the higher rates of multimorbidities that they experience compared with other people, and amenable deaths.[34]

While accommodation type (not living with a family carer), ability level, not having day-time occupation, having spastic quadriplegia, visual impairment, constipation, ataxia/gait disorder, osteoporosis, hypertension, dysphagia, dyspnoea, gastro-oesophageal reflux disorder, total number of physical health disorders, not having impaired mobility, not having urinary incontinence and not having autism, number of general practitioner consultations in the previous 12 months, total number of different types of health professionals providing care at the time of the health assessment and antiepileptic drugs were related to time of death on univariate analyses, they were not retained in the multivariable model.

The majority of the adults with intellectual disabilities, with and without Down syndrome, died in an NHS hospital.

### Comparison with previous literature

The overall SMR we report, higher SMR in women than men and higher SMR at younger age groups is similar to the majority of previous reports. Most mortality studies with people with Down syndrome have been

conducted with children. Previous reports of children and adults (combined) gave an SMR=5.5,[20] and for adults SMR=7.6,[9] compared with our finding for adults with Down syndrome of SMR=5.28. Recent systematic reviews reported people with intellectual disabilities on average died 20 years younger than other people, and people with Down syndrome died 28 years younger, although the majority of the Down syndrome studies were not recent.[1 2] In our study, we found the gap between the age at death of people with intellectual disabilities with and without Down syndrome to be only 5.3 years, possibly reflecting the increasing lifespan of people with Down syndrome exceeding increases in lifespan for people with intellectual disabilities without Down syndrome. Notably, after 'Down syndrome', dementia was the most commonly reported underlying, and all contributing cause of death for the adults with Down syndrome, whereas studies in the past commented on congenital heart disease and respiratory causes.

For the cohort overall, respiratory infection and aspiration/reflux/choking were the most common all contributing causes of death. These conditions feature in previous studies on causes of death,[5 6 8 10 11] although there are inconsistencies between studies. By ICD-10 chapter, our study found the most common underlying causes of death were diseases of the respiratory system, then of the circulatory system, followed by neoplasms. Others reported the most common to be vascular,[10] circulatory,[5] heart disease[17] and jointly circulatory and neoplasm.[19]

Previous research from other countries has highlighted that listing Down syndrome or intellectual disabilities as the underlying cause of death obscures actual causes of death for this population.[35] We therefore presented data on revised cause of death for the 21 people for whom it was listed as Down syndrome (as a sensitivity check), and highlight with interest that in this Scottish cohort, no one had intellectual disabilities listed as underlying cause of death. This may reflect different medical death certificate recording practices in Scotland compared with for example, the USA.

Studies that investigated avoidable deaths in adults with intellectual disabilities found them to be more common than in the general population, due to deaths that would have been amenable to good care. Avoidable deaths have been reported in 44.7% of deaths of people with intellectual disabilities in England (mostly amenable deaths—figure not reported),[6] and in 31% in Australia,[19] compared with our figure of 38.9%. Avoidable deaths that would have been amenable to good care have been reported to occur in 37% of deaths of people with intellectual disabilities in England.[5] Our figure is slightly lower at 29.8% but still more than double that found in the Scottish general population.[31] It should be noted that the ONS list of avoidable deaths was not designed specifically for people with intellectual disabilities, and it may emphasise some causes less relevant, and omit others that might be highly relevant in this population.[5]

## Strengths and limitations

The strengths of the study include the thorough methods of case ascertainment for intellectual disabilities at baseline with verification of intellectual disabilities and its severity, suggesting results are generalisable in other high-income countries. While our identification of the population will not have identified everyone with intellectual impairment (an IQ<70), in view of the multiple sources used we believe it will have identified the adults with intellectual disabilities (IQ<70, plus need for support in daily activities, and onset in the developmental period). Additionally, there were detailed clinical assessments at baseline, and a longitudinal design. The size of the cohort and the duration of follow-up is also a strength, as is the successful record linkage for 94% of participants. Our study does have limitations, specifically that the study was only conducted in one region of Scotland, and the reliance on death certificate data to obtain cause of death. Additionally, the characteristics and health of the participants was collected in 2002–2004. The health conditions we investigated tend to be long-standing or remitting/relapsing conditions, and psychotropic prescribing also once initiated tends to be long-standing in people with intellectual disabilities. However, it is possible that extent of neighbourhood deprivation, type of accommodation, employment and civil status (though few marry) might have changed for some people between 2002 and 2004 and 2018; we have no further information to check this. There were no concerns regarding the proportional hazards assumption in the multivariable model. The linkage was also reliant on the accuracy of the CHI number as a sole source of linkage.

## Implications

It is important to know the factors that are associated with risk of death, and the common causes of death in this population, as these then inform the actions needed to reduce the unacceptably high SMRs experienced by people with intellectual disabilities. Awareness of these factors may provide a pathway to action to reduce the observed earlier mortality in adults with intellectual disabilities. It is not adequate to solely rely on the public health interventions available to everyone, even when they are accessible. Aspiration, reflux and choking could, and should, be avoided by raising awareness of its consequences (death), and putting in place training on simple measures related to feeding, positioning, food consistency, and when to seek health advice from speech and language therapy, physiotherapy, nursing, and medical advice. Carers need to be aware of how the adults they care for express pain, so that conditions such as gastrointestinal ulcers are attended to, prior to the extreme point of perforation and so treatable conditions such as constipation and urinary tract infections are managed before they lead to respiratory distress and sepsis. Quality of care is important; adults with intellectual disabilities need just as good care for their diabetes and epilepsy (and other conditions) as the rest of the population, with

reasonable adjustments to address accessibility, and accessible smoking cessation programmes.

## Future research

Further research on larger samples is needed, particularly with regard to replicating and extending our findings on the factors that are associated with risk of death, and any sex differences in them, so that practitioners can focus on actions to improve the life expectancy of adults with intellectual disabilities, with and without Down syndrome.

**Acknowledgements** We thank the NHS Greater Glasgow and Clyde learning disabilities primary care liaison team for their contribution to the study, and the participants with intellectual disabilities and their carers.

**Contributors** S-AC is principle investigator, she conceived and managed the project, interpreted data and wrote the first draft of the manuscript. LA contributed to the conception of the project, and project management. NG designed and supervised the statistical analysis, and contributed to data interpretation and drafting of the manuscript. PMcS implemented and refined the statistical analysis, and contributed to data interpretation and drafting of the manuscript. AJ implemented and refined the statistical analysis, and contributed to data interpretation. AH contributed to data linkage and interpretation, and drafting of the manuscript. CMcC provided expertise on data linkage and methods, and drafting of the manuscript. DK contributed to data interpretation and drafting of the manuscript. CM contributed to data interpretation, and drafting of the manuscript. All approved the final version of the manuscript. S-AC is the study guarantor.

**Funding** This work was supported by the UK Medical Research Council, grant number: MC_PC_17217), and the Scottish Government via the Scottish Learning Disabilities Observatory. The study sponsor and funders had no role in the study design; in the collection, analysis and interpretation of data; in the writing of the report; and in the decision to submit the article for publication. The researchers are independent from the funders.

**Competing interests** None declared.

**Patient consent for publication** Not required.

**Ethics approval** Ethical approval was gained from NHS Greater Glasgow Primary Care Trust- Community & Mental Health Research Ethics Committee, and NHS Greater Glasgow and Clyde Safe Haven. Individual consent to participate was taken in line with Scottish law, between 2001–2004.

**Provenance and peer review** Not commissioned; externally peer reviewed.

**Data availability statement** Data may be obtained from a third party and are not publicly available. Data are available via NHS Greater Glasgow & Clyde safe haven upon application.

**ORCID iDs**
Sally-Ann Cooper http://orcid.org/0000-0001-6054-7700
Angela Henderson http://orcid.org/0000-0002-6146-3477

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
