## [Reviewer comments · BMJ Open]

ARTICLE DETAILS

TITLE (PROVISIONAL)	Rates, causes, place, and predictors of mortality in adults with intellectual disabilities with and without Down syndrome: cohort study with record linkage
AUTHORS	Cooper, Sally-Ann; Allan, Linda; Greenlaw, Nicola; Mcskimming, Paula; Jasilek, Adam; Henderson, Angela; McCowan, Colin; Kinnear, Deborah; Melville, Craig

VERSION 1 – REVIEW

REVIEWER	Scott Landes Syracuse University, USA
REVIEW RETURNED	13-Jan-2020

GENERAL COMMENTS	This paper addresses the under-researched topic of mortality trends for adults with intellectual disability, a population that dies at ages 20-25 years earlier than their peers in the general population. As such, it provides needed empirical evidence that can prove informative to general medical practitioners providing care for adults with intellectual disability, as well as to the disability research field. The authors are to be commended for their study of this topic. I have some suggestions, in no certain order, that I think would improve the clarity and thoroughness of the results. 1. It appears, per the reference on page 7, that the characteristics utilized to predict age to death in the univariate and multivariate Cox models are based upon baseline measurement for all study participants in 2002-2004. If this is indeed the case, I think it presents some challenges to these models. If this is not when these characteristics were measured, this needs to be clarified. Assuming the predictors in Table 6 and Supplementary Table 3 are baseline measurements, I have concern that many of these are time variant in ways that could influence mortality outcomes. The obvious characteristics that meet this definition would be diagnoses such as incontinence, respiratory infection, prescription medications, etc. Each of these could be present at baseline, but not during the duration of the study, or, could not be present at baseline, but later acquired prior to the end of the study. As such, I am not sure if they are accurate predictors of time to death. This concern extends beyond the physiological characteristics to other predictors, such as type of accommodation and neighborhood deprivation, that also may have changed across the study and likely influence mortality risk. As Cox models assume proportional hazards across the study period, the presence of a time variant diagnosis at baseline may violate this assumption. Using static characteristics such as sex, ability level, DS, and chronic diagnoses such as epilepsy, quadriplegia, hearing and visual impairment, any mental illness, etc. makes sense in this
--

analysis as there is greater surety that these predictors endure across the time of the study. However, I am concerned that the use of time variant predictors in these models that are so strongly associated with mortality risk may be misleading, and may violate the proportional hazards assumption. I really wonder if this is why bowel incontinence is shown to be associated with reduced mortality risk. While this does not negate the use of Cox models, supplementary analysis that tests the proportional hazards assumption should be conducted and reported.

2. Based upon prior research that reports distinct differences in mortality risk for female and males, and across the life course, I think it would be best if all models in Supplementary table 3 were adjusted for age and biological sex.

3. I would like to see the age categories and biological sex included in the multivariate Cox model reported in Table 6. From a demographic point of view, sex should always be included in models, but I think especially here as you have earlier indicated differences by sex. Use of age categories could determine whether your results replicate, or not, earlier studies reporting a decrease in the intellectual mortality disadvantage over time.

4. I think it would be helpful to provide more information on the amenable and avoidable deaths. Examples of each could be provided on page 10, then some indication of the specific avoidable and amenable deaths present in this study in the results. I think this would be extremely helpful in helping medical practitioners focus on preventable causes of death in this population. I really find this a key discussion in this paper and would like to see the specific results.

5. I was somewhat perplexed by the groupings of causes of death provided in Supplementary Table 2. The majority of these make logical sense, but a few of interest need further clarification. I think one addition that would prove extremely helpful across the board would be to add the ICD10 codes for each condition. This would help readers contextualize findings within broader results. Back to my main point. My most pressing question regarding the grouping of aspiration/reflux/choking into one category. In many ways this makes sense, and I think one of the strongest contributions of this paper is highlighting the high rates of death from these preventable causes. However, although very much related, pneumonitis (classified as a respiratory disease in ICD10) and the choking codes (classified as an external cause in ICD10) are distinct codes. While I do not think these necessarily need to be disaggregated in the paper, there should be sufficient explanation that this category, as well as others in the list, include combinations of diagnoses from different chapters, and the rationale for doing so. I do not disagree with a grouping such as the one I detail, but think more information should be provided to ensure readers understand the logic for these decisions.

6. At the least, some discussion is warranted regarding the coding of developmental disabilities (intellectual disability, Down syndrome, cerebral palsy) as underlying cause of death. Prior research demonstrates that this practice obscures actual causes of death for this population, which is not at all helpful for public health efforts, especially as many of the obscured causes of death are highly preventable. As this problem is well-documented, it is necessary to discuss why the decision was made not to revise these death certificates, and the possible effect of including developmental disabilities as COD.

7. Finally, this paper address a wide array of data surrounding mortality risk for adults with intellectual disability: demographic and health predictors; age and biological sex differences; specific cause

	and multiple cause trends; preventable causes. It would be helpful to provide a more integrative narrative at the front and back end of the paper that frames these findings. As stated at the beginning of my review, this is an under-researched topic, and the results from this paper have the potential to provide needed insight into mortality risk for adults with intellectual disability. I think providing a more nuanced framework for the paper that helps integrate the important findings in such a way that is more focused on a cohesive narrative would make a more compelling presentation for medical practitioners.
--	--

REVIEWER	Emily Lauer University of Massachusetts Medical School, USA
REVIEW RETURNED	23-Jan-2020

GENERAL COMMENTS	Overall, this is a high quality manuscript representing a well done study that is an important contribution to the literature. Minor revisions are requested as follows. The abstract for this paper is choppy and hard to follow. Please review and revise for clarity. On page 5, lines 48 - 56, it's important to note in the study comparison that the studies used different levels of groupings of causes of death (e.g. pneumonia, vs. respiratory system), which can meaningfully affect rankings between studies. Page 6 sentence starting at line 9 - should this state children instead of people? It is an important distinction whether these are the most common causes for people of any age vs. only children. Page 8 - line 44 - I believe this should state "immediate causes of death" Page 10 line 46, please review this sentence for clarity In the groupings of causes of death for people with Down Syndrome, I would encourage the authors to consider how the groupings may change if the cause of "Down Syndrome" was ignored as an underlying cause of death. This analysis could at least be conducted as a sensitivity analysis, and also brings important consideration when examining potential preventable and amenable deaths for this subgroup. There are known issues with diagnostic overshadowing in underlying causes of death for people with ID such that their ID is listed as an underlying cause inaccurately. The fact that Down Syndrome was the most commonly listed underlying cause of death is very likely masking important patterns in actual underlying causes that would permit more direct comparison with the ID population in this study. For limitations of the study, despite the various methods used to identify people, there is still a risk that people with ID in the target area were missed, and that these people may have real differences from the identified cohort. For example, they would have been less likely to be in receipt of medical care, and likely would have had more mild ID (therefore less readily recognized and less likely to seek services). The linkage was also reliant on the accuracy of the connected identifier number as a sole source of linkage. Supp Table 1: The SMR data for the USA states that it is not reported. Please consider this source as an example of US SMR data. Lauer E & McCallion P. Mortality of People with Intellectual
--

	and Developmental Disabilities from Select US State Disability Service Systems and Medical Claims Data. Journal of Applied Research in Intellectual Disabilities 2015. doi 10.1111/jar.12191
--	--

VERSION 1 – AUTHOR RESPONSE

Reviewer: 1

Reviewer Name: Scott Landes

Institution and Country: Syracuse University, USA

Please state any competing interests or state 'None declared': None declared

This paper addresses the under-researched topic of mortality trends for adults with intellectual disability, a population that dies at ages 20-25 years earlier than their peers in the general population. As such, it provides needed empirical evidence that can prove informative to general medical practitioners providing care for adults with intellectual disability, as well as to the disability research field. The authors are to be commended for their study of this topic.

I have some suggestions, in no certain order, that I think would improve the clarity and thoroughness of the results.

1. It appears, per the reference on page 7, that the characteristics utilized to predict age to death in the univariate and multivariate Cox models are based upon baseline measurement for all study participants in 2002-2004. If this is indeed the case, I think it presents some challenges to these models. If this is not when these characteristics were measured, this needs to be clarified. Assuming the predictors in Table 6 and Supplementary Table 3 are baseline measurements, I have concern that many of these are time variant in ways that could influence mortality outcomes. The obvious characteristics that meet this definition would be diagnoses such as incontinence, respiratory infection, prescription medications, etc. Each of these could be present at baseline, but not during the duration of the study, or, could not be present at baseline, but later acquired prior to the end of the study. As such, I am not sure if they are accurate predictors of time to death. This concern extends beyond the physiological characteristics to other predictors, such as type of accommodation and neighborhood deprivation, that also may have changed across the study and likely influence mortality risk. As Cox models assume proportional hazards across the study period, the presence of a time variant diagnosis at baseline may violate this assumption. Using static characteristics such as sex, ability level, DS, and chronic diagnoses such as epilepsy, quadriplegia, hearing and visual impairment, any mental illness, etc. makes sense in this analysis as there is greater surety that these predictors endure across the time of the study. However, I am concerned that the use of time variant predictors in these models that are so strongly associated with mortality risk may be misleading, and may violate the proportional hazards assumption. I really wonder if this is why bowel incontinence is shown to be associated with reduced mortality risk. While this does not negate the use of Cox models, supplementary analysis that tests the proportional hazards assumption should be conducted and reported.

***Response:

It is correct that the participant characteristics included in the Cox models were collected in 2002-2004, when all the participants had a detailed health assessment. This includes age at the time of the health assessment, which was the measure of age that was included in these models. We also had age at death, which has been summarised separately for the “age at death” for this population. The two medically qualified authors have reviewed the 34 health conditions investigated, and consider that, other than respiratory infection, they do tend to be persistent or remitting/relapsing in this population with intellectual disabilities, and typically require long-term management. Even lower respiratory tract infection may tell us something about the persons who had it; a vulnerability. Some

individuals are prone to repeated lower respiratory track infections, and whilst we do not know to which of the participants this applied, we note that lower respiratory track infection was retained in the multivariable analyses as related to mortality, and was the most common all contributing cause of death. Regarding the 5 drug items, once psychotropic medications are prescribed for people with intellectual disabilities, they tend to be long-standing prescriptions; indeed there is considerable concern in the UK about this, and campaigns such as “STOMP” to try to reduce the long-term prescribing of these drugs. Polypharmacy is also a recognised problem in this population. The 3 items on extent of service use highlights information about the participants health during 2002-2004. Of the 8 demographic items, we agree that neighbourhood deprivation, type of accommodation, employment, and civil status (though few marry) might have changed for some people between 2002-2004 and 2018, but feel it important to include these items as information on their life situations in 2002-2004. Our team’s senior statistician (NG) also checked the statistical assumptions of the multivariable model, and there are no concerns regarding the proportional hazards assumption. We have added as a limitation in the discussion (page 21):

“The characteristics and health of the participants was collected in 2002-2004. The health conditions we investigated tend to be long-standing or remitting/relapsing conditions, and psychotropic prescribing also once initiated tends to be long-standing in people with intellectual disabilities. However, it is possible that extent of neighbourhood deprivation, type of accommodation, employment, and civil status (though few marry) might have changed for some people between 2002-2004 and 2018; we have no further information to check this. There were no concerns regarding the proportional hazards assumption in the multivariable model”

2. Based upon prior research that reports distinct differences in mortality risk for female and males, and across the life course, I think it would be best if all models in Supplementary table 3 were adjusted for age and biological sex.

***Response:

Most of the prior published research does not show a difference in mortality risk between men with intellectual disabilities and women with intellectual disabilities. It shows a greater difference for women with intellectual disabilities compared with women without intellectual disabilities, than is the difference for men with intellectual disabilities compared with men without intellectual disabilities (the differences in male and female SMRs are because in the general population women live longer than men, not any difference in the intellectual disabilities population between men and women). Age and sex were both used in the multivariable analysis which is the main analysis included in the paper. The multivariable model is stepwise, and statistically, sex was not significantly required for the model, hence is not included in table 6. The univariate analyses are, by definition, not adjusted. Age and sex are reported individually as univariate results along with the other variables in supplementary table 3. Having carefully considered the reviewer’s point, we prefer to leave the univariate analyses as they are in supplementary table 3, and have not added an additional supplementary table adjusting for age and sex, although will do so if the editor wishes us to.

3. I would like to see the age categories and biological sex included in the multivariate Cox model reported in Table 6. From a demographic point of view, sex should always be included in models, but I think especially here as you have earlier indicated differences by sex. Use of age categories could determine whether your results replicate, or not, earlier studies reporting a decrease in the intellectual mortality disadvantage over time.

***Response:

Variables for sex and age were permitted in to the multivariable stepwise model. Sex was not retained in the model, as it did not meet statistical significance, both in the univariate and the multivariable

models for time to death. Age at the time of the health assessment was included in the model as a continuous measure, as that provides more statistical power than using a number of age categories. We have rewritten the description of this in the methods so that it is clearer (page 11):
“All 50 variables were then permitted entry in to a single multivariable analysis using stepwise regression methods, in order to identify a model containing the statistically significant factors associated with death. Age at date of the health assessment was entered in to the model as a continuous measure.”

In table 2, we do report SMR by age categories, which allows comparison with previous studies, and shows similarities.

4. I think it would be helpful to provide more information on the amenable and avoidable deaths. Examples of each could be provided on page 10, then some indication of the specific avoidable and amenable deaths present in this study in the results. I think this would be extremely helpful in helping medical practitioners focus on preventable causes of death in this population. I really find this a key discussion in this paper and would like to see the specific results.

***Response:

As well as providing the reference to the full list of deaths that ONS classify as avoidable, we have now added examples in the methods to clarify these terms (page 10):
“The Office for National Statistics (ONS) published a definition of avoidable mortality,³⁰ which lists the causes of amenable deaths (deaths that should not occur in the presence of good health care, e.g. respiratory disease), and causes of preventable deaths (e.g. from diseases that could have been avoided by prior immunisation) by ICD-10 codes.”

The causes of avoidable deaths are included in table 5. Some causes have necessarily had to be grouped due to small numbers and the risk of statistical disclosure, and for this reason we are not really able to generate a separate table on the avoidable deaths. Supplementary table 2 also includes all of the causes of deaths in the cohort (but without the frequency data), so all avoidable deaths are listed in there. We have amended this sentence in the results to guide readers to this:
“According to the ONS list of avoidable deaths, 102 (38.9%) of the 262 deaths were avoidable; most notably, respiratory infection and epilepsies (table 4).”

Additionally, we make the comment in the discussion on the limitations of the list of avoidable deaths for adults with intellectual disabilities:
“The proportion of deaths that would have been amenable to good care for adults with intellectual disabilities was more than double that seen in the general population. Although aspiration/reflux/choking is not included in the ONS list of avoidable deaths, and therefore not included in the figures we report on amenable deaths, we consider that good care could have prevented many of these deaths. This appears to be very important for adults with intellectual disabilities irrespective of whether they have Down syndrome. Similarly, some other causes of deaths within this cohort (supplementary table 2), such as constipation/mega-colon, and urinary tract infections do not appear on the ONS list of avoidable deaths.”

5. I was somewhat perplexed by the groupings of causes of death provided in Supplementary Table 2. The majority of these make logical sense, but a few of interest need further clarification. I think one addition that would prove extremely helpful across the board would be to add the ICD10 codes for each condition. This would help readers contextualize findings within broader results. Back to my main point. My most pressing question regarding the grouping of aspiration/reflux/choking into one category. In many ways this makes sense, and I think one of the strongest contributions of this paper

is highlighting the high rates of death from these preventable causes. However, although very much related, pneumonitis (classified as a respiratory disease in ICD10) and the choking codes (classified as an external cause in ICD10) are distinct codes. While I do not think these necessarily need to be disaggregated in the paper, there should be sufficient explanation that this category, as well as others in the list, include combinations of diagnoses from different chapters, and the rationale for doing so. I do not disagree with a grouping such as the one I detail, but think more information should be provided to ensure readers understand the logic for these decisions.

***Response:

The ICD-10 codes have been added to supplementary table 2.

We have rewritten the following paragraph for greater clarity (page 8/9):

“In order to provide finer granularity of cause of death, two clinical academics then grouped specific causes of death into narrower groupings than those provided by ICD-10 chapter headings (supplementary table 2). This approach was also in view of the recognised issue of variation between health staff in distinguishing and recording immediate causes of death, and because some causes occurred in low numbers so could not be individually reported due to the risk of statistical disclosure. Additionally, some conditions likely to be the same are split between different ICD-10 chapters, e.g. dementia in Alzheimer disease (F00) and unspecified dementia (F03) in the mental and behavioural disorders chapter, and Alzheimer’s disease (G30) and Alzheimer’s disease, unspecified (G30.9) in the diseases of the nervous system chapter. A list of related conditions was generated by one of the clinical academics and then checked by the second.”

6. At the least, some discussion is warranted regarding the coding of developmental disabilities (intellectual disability, Down syndrome, cerebral palsy) as underlying cause of death. Prior research demonstrates that this practice obscures actual causes of death for this population, which is not at all helpful for public health efforts, especially as many of the obscured causes of death are highly preventable. As this problem is well-documented, it is necessary to discuss why the decision was made not to revise these death certificates, and the possible effect of including developmental disabilities as COD.

***Response:

We have added information after reclassifying the underlying cause of death for the 21 people for whom it was recorded on the death certificate as Down syndrome (page 14) as follows

“For the 21 people whose death certificate listed Down syndrome as their underlying cause of death, the death certificates were reviewed and underlying cause of death reclassified, as a sensitivity check. Following this, the most common underlying causes of death for the adults with Down syndrome were dementia (n=20; 35.1%), then other infection (n=7; 12.3%).”

None of the adults had intellectual disabilities recorded on their death certificates as their underlying cause of death. The reviewer has prompted us to reflect that this differs to data from the USA, and so is an important finding, hence we have added the following to the discussion (page 20):

“Previous research from other countries has highlighted that listing Down syndrome or intellectual disabilities as the underlying cause of death obscures actual causes of death for this population.³⁴ We therefore presented data on revised cause of death for the 21 people for whom it was listed as Down syndrome (as a sensitivity check), and highlight with interest that in this Scottish cohort no-one had intellectual disabilities listed as underlying cause of death. This may reflect different medical recording practices in Scotland compared to e.g. USA.”

34. Landes SD, Stevens JD, Turk MA. The obscuring effect of coding developmental disability as the underlying cause of death on mortality trends for adults with developmental disability: a cross-sectional study utilizing U.S. mortality data from 2012 to 2016. *BMJ Open*, 2019. 9:e026614.

7. Finally, this paper address a wide array of data surrounding mortality risk for adults with intellectual disability: demographic and health predictors; age and biological sex differences; specific cause and multiple cause trends; preventable causes. It would be helpful to provide a more integrative narrative at the front and back end of the paper that frames these findings. As stated at the beginning of my review, this is an under-researched topic, and the results from this paper have the potential to provide needed insight into mortality risk for adults with intellectual disability. I think providing a more nuanced framework for the paper that helps integrate the important findings in such a way that is more focused on a cohesive narrative would make a more compelling presentation for medical practitioners.

***Response

At the end of the first paragraph of the introduction, we have now added (page 4):

“There has been a recent increase in research on mortality in people with intellectual disabilities, but very little research has distinguished people with intellectual disabilities with and without Down syndrome, or investigated the factors associated with risk of mortality, and causes of mortality.”

Prior to commenting on specific examples, in the “implications” section of the discussion (page 22) we have added:

“Awareness of these factors may provide a pathway to action to reduce the observed earlier mortality in adults with intellectual disabilities.”

Reviewer: 2

Reviewer Name: Emily Lauer

Institution and Country: University of Massachusetts Medical School, USA

Please state any competing interests or state ‘None declared’: None

Overall, this is a high quality manuscript representing a well done study that is an important contribution to the literature. Minor revisions are requested as follows.

The abstract for this paper is choppy and hard to follow. Please review and revise for clarity.

***Response:

We agree with the reviewer, but the abstract is currently exactly 300 words long, which is the limit allowed by the journal. To improve the flow in it, we would need to reduce the amount of results reported, which we prefer not to do, given that abstracts are the most read part of a paper.

On page 5, lines 48 - 56, it's important to note in the study comparison that the studies used different levels of groupings of causes of death (e.g. pneumonia, vs. respiratory system), which can meaningfully affect rankings between studies.

***Response:

We agree and have added the following (pages 5/6):

“Additionally, studies group causes of death in different ways (e.g. pneumonia versus respiratory system), which can affect prevalence rankings between studies.”

Page 6 sentence starting at line 9 - should this state children instead of people? It is an important distinction whether these are the most common causes for people of any age vs. only children.

***Response:

We have now corrected this error as follows (page 6):

"In people with Down syndrome, most studies on mortality have been conducted with child populations, and report the most common causes of death to be congenital heart disease, and pneumonia/diseases of the respiratory system.²"

Page 8 - line 44 - I believe this should state "immediate causes of death"

***Response:

We have changed "individual" to "specific" to improve clarification (page 8).

Page 10 line 46, please review this sentence for clarity

***Response:

We have rewritten the sentence for clarity, as follows (page 11):

"All 50 variables were then permitted entry in to a single multivariable analysis using stepwise regression methods, in order to identify a model containing the statistically significant factors associated with death. Age at date of the health assessment was entered in to the model as a continuous measure."

In the groupings of causes of death for people with Down Syndrome, I would encourage the authors to consider how the groupings may change if the cause of "Down Syndrome" was ignored as an underlying cause of death. This analysis could at least be conducted as a sensitivity analysis, and also brings important consideration when examining potential preventable and amenable deaths for this subgroup. There are known issues with diagnostic overshadowing in underlying causes of death for people with ID such that their ID is listed as an underlying cause inaccurately. The fact that Down Syndrome was the most commonly listed underlying cause of death is very likely masking important patterns in actual underlying causes that would permit more direct comparison with the ID population in this study.

***Response:

We agree and have added information after reclassifying the underlying cause of death for the 21 people for whom it was recorded on the death certificate as Down syndrome (page 14) as follows:

"For the 21 people whose death certificate listed Down syndrome as their underlying cause of death, the death certificates were reviewed and underlying cause of death reclassified, as a sensitivity check. Following this, the most common underlying causes of death for the adults with Down syndrome were dementia (n=20; 35.1%), then other infection (n=7; 12.3%)."

We have also added (Page 20):

"Previous research from other countries has highlighted that listing Down syndrome or intellectual disabilities as the underlying cause of death obscures actual causes of death for this population.³⁴ We therefore presented data on revised cause of death for the 21 people for whom it was listed as Down syndrome (as a sensitivity check), and highlight with interest that in this Scottish cohort no-one had intellectual disabilities listed as underlying cause of death. This may reflect different medical death certificate recording practices in Scotland compared to e.g. USA."

For limitations of the study, despite the various methods used to identify people, there is still a risk that people with ID in the target area were missed, and that these people may have real differences

from the identified cohort. For example, they would have been less likely to be in receipt of medical care, and likely would have had more mild ID (therefore less readily recognized and less likely to seek services). The linkage was also reliant on the accuracy of the connected identifier number as a sole source of linkage.

***Response:

We make the distinction between intellectual impairment (IQ<70) and intellectual disabilities (IQ<70 plus need for support in daily activities). We certainly will not have identified everyone with intellectual impairment, but believe we will have missed few people with intellectual disabilities as the identification process included multiple sources, including people receiving financial support for any services for intellectual disabilities, and all general practitioners, and not just the intellectual disabilities service. We have added the following to the strengths and limitations section of the discussion (page 21):

“Whist our identification of the population will not have identified everyone with intellectual impairment (an IQ<70), in view of the multiple sources used, we believe it will have identified the adults with intellectual disabilities (IQ<70, plus need for support in daily activities, and onset in the developmental period).”

We have also added to the limitations (page 21):

“The linkage was also reliant on the accuracy of the CHI number as a sole source of linkage.”

Supp Table 1: The SMR data for the USA states that it is not reported. Please consider this source as an example of US SMR data. Lauer E & McCallion P. Mortality of People with Intellectual and Developmental Disabilities from Select US State Disability Service Systems and Medical Claims Data. Journal of Applied Research in Intellectual Disabilities 2015. doi 10.1111/jar.12191

***Response:

We have updated the supplementary table 1 with data from this important paper.

VERSION 2 – REVIEW

REVIEWER	Scott Landes Syracuse University, USA
REVIEW RETURNED	17-Apr-2020

GENERAL COMMENTS	Excellent revision. All my suggestions/concerns were sufficiently addressed. This is a solid paper that I look forward to seeing in print. I have one minor suggestion to tie the results to an earlier study. In the discussion section you state "SMRs were lowest with older age groups, likely to be due to increased illness in the older general population and conversely a healthier group with intellectual disabilities living to older ages compared with those who die younger." This confirms a finding in an earlier study (Landes, Scott D. 2017. "The Intellectual Disability Mortality Disadvantage: Diminishing with Age?". American Journal on Intellectual and Developmental Disabilities 122(2):192-207.) in which I provide a very similar interpretation of heterogeneity of frailty. I do think it worthwhile to expand this explanation a bit to ensure the reader understands the finding - more or less, if able to survive into older adulthood, individuals with intellectual disability are nearly as, or just as, hardy as those without disability.
---

VERSION 2 – AUTHOR RESPONSE

Reviewer: 1

Reviewer Name: Scott Landes

Institution and Country: Syracuse University, USA

Please state any competing interests or state 'None declared': None

Excellent revision. All my suggestions/concerns were sufficiently addressed. This is a solid paper that I look forward to seeing in print.

I have one minor suggestion to tie the results to an earlier study. In the discussion section you state "SMRs were lowest with older age groups, likely to be due to increased illness in the older general population and conversely a healthier group with intellectual disabilities living to older ages compared with those who die younger." This confirms a finding in an earlier study (Landes, Scott D. 2017. "The Intellectual Disability Mortality Disadvantage: Diminishing with Age?". *American Journal on Intellectual and Developmental Disabilities* 122(2):192-207.) in which I provide a very similar interpretation of heterogeneity of frailty. I do think it worthwhile to expand this explanation a bit to ensure the reader understands the finding - more or less, if able to survive into older adulthood, individuals with intellectual disability are nearly as, or just as, hardy as those without disability.

***Response:

We have added the reference to the end of this sentence in the discussion (page 18):

SMRs were lowest with older age groups, likely to be due to increased illness in the older general population and conversely a healthier group with intellectual disabilities living to older ages compared with those who die younger (as has previously been reported³⁴).